# Exploring Volatiles and Biological Effects of *Commiphora africana* and *Boswellia papyrifera* Incense

**DOI:** 10.3390/molecules30030499

**Published:** 2025-01-23

**Authors:** Sara A. Eltigani, Chisato Ohta, Ryota Nakamiya, Mizuki Yokono, Tomohiro Bito, Kenji Takahashi, Yukinori Yabuta, Mohamed M. Eltayeb, Toshio Ohta, Atsushi Ishihara

**Affiliations:** 1School of Health Sciences, Ahfad University for Women, Arda Street, Omdurman 14411, Sudan; saraahmed8611@gmail.com; 2Department of Agricultural, Life, Environmental Sciences, Faculty of Agriculture, Tottori University, Tottori 680-8553, Japan; d.best121106@gmail.com (C.O.); ryotartm@gmail.com (R.N.); bito@tottori-u.ac.jp (T.B.); yabuta@tottori-u.ac.jp (Y.Y.); 3Technical Department, Tottori University, Tottori 680-8550, Japan; yokono@tottori-u.ac.jp; 4Joint Department of Veterinary Medicine, Faculty of Agriculture, Tottori University, Tottori 680-8553, Japan; takahashi-k@tottori-u.ac.jp (K.T.); tohta@tottori-u.ac.jp (T.O.); 5Department of Food Science and Technology, Faculty of Agriculture, University of Khartoum, Khartoum North 14413, Sudan; meltayeb@tottori-u.ac.jp; 6International Platform for Dryland Research and Education, Tottori University, Tottori 680-0001, Japan

**Keywords:** *Commiphora africana*, *Boswellia papyrifera*, volatile, tyrosinase inhibitor, cancerous cells, *Streptococcus mutans*, Sudanese fragrant

## Abstract

The resin of *Commiphora africana* and the resin and bark of *Boswellia papyrifera* play versatile roles in traditional Sudanese culture, including use in inhalation therapy, liquid remedies, and as chewing gum. Thus, this study aimed to analyze the volatile compounds in these materials using various extraction methods and assess their biological activities. Extraction methods included MonoTrap solid-phase microextraction, smoke solvent trapping, and acetone immersion. Gas chromatography−mass spectrometry analysis of MonoTrap extracts identified highly volatile compounds, while smoke extracts contained compounds with lower volatility. Solvent immersion captured a broader range of compounds. The resin of *C. africana* was rich in limonene, verbenone, and β-selinene, whereas *B. papyrifera* extracts contained octyl acetate, trans-nerolidol, and nerolidol isobutyrate as major compounds. Biological assays showed *C. africana* smoke extract inhibited tyrosinase activity, with *p*-cymene and *S*-limonene acting as competitive inhibitors. It also inhibited the growth of cancer cells, A549 and MIA Paca-2, while solvent extracts from both resins inhibited all tested cell lines. Further, the acetone extracts exhibited strong antibacterial activity against *Streptococcus mutans*. These results highlight the differences in chemical composition between the two species, the impact of extraction methods, and the therapeutic potential of *C. africana* and *B. papyrifera* as sources of bioactive compounds.

## 1. Introduction

Medicinal plants have been used for centuries around the world to treat a variety of diseases [1,2]. These plants are administered in numerous ways, such as swallowing, drinking, inhalation, and other external applications. In African traditions, inhalation therapy is widely practiced, where plant or animal materials are burned, and the emitted smoke is inhaled to treat ailments, including respiratory, dermal, and depression-related disorders [3,4]. In Sudan, this inhalation therapy, known as Bakhoor therapy, involves burning animal and plant materials to produce a sweet-smelling smoke that is believed to have therapeutic and aesthetic benefits [5].

In Sudan, various plant materials are used in Bakhoor therapy, with the bark of Boswellia spp., locally called Tarq-Tarq, being one of the most popular materials. Additionally, resins from both *Commiphora* and *Boswellia* species are commonly used. Both genera belong to the Burseraceae family, which includes over 150 species of trees and shrubs found throughout East Africa, Arabia, and India [6,7]. The resins of Boswellia spp., known as frankincense, are used worldwide, with the main frankincense-producing species in Africa being *Boswellia papyrifera* (Delile ex Caill.) Hochst., which is found in Ethiopia and Sudan [8]. This shrub grows in Central and Southern Sudan, and its resin is imported from Saudi Arabia and North Africa. It is chewed and used in Bakhoor. *Commiphora* spp. is a small shrub commonly found in Western and Eastern Sudan. The most widely used *Commiphora* spp. in Sudan is *Commiphora africana* (A.Rich.) Endl. Its stem is used to treat rheumatic diseases, and its resin (myrrh) is chewed and burned to produce smoke for treating wounds, gingivitis, swellings, flatulence, colic, digestive issues, and antispasmodic conditions [9]. Myrrh is also used to treat a wide range of conditions, including ulcerative colitis, fever, gallbladder issues, skin infections, dysmenorrhea, amenorrhea, tumors, and chest ailments [10,11,12,13]. Additionally, it is used for skin nourishment [14]. Frankincense and myrrh can be burned together or with other herbs to produce pleasant smoke that is used to treat respiratory issues and alleviate phlegm.

Studies of *B. papyrifera* resin have revealed that it contains high levels of octyl acetate, octanol, and bornyl acetate, as well as varying amounts of diterpenoids [15]. In addition, the resins of *C. africana* and *B. papyrifera* have been demonstrated to hold multiple aromatic compounds, including monoterpenes and sesquiterpenes like α-pinene, α-thujene, sabinene, limonene, myrcene, *p*-cymene, and β-caryophyllene [15,16]. Further, a tricyclic triterpene acid, commafric A, has been reported as a characteristic component in *C. africana* resin [16]. These compounds are known for their antibacterial, anticancer, and anti-inflammatory activities [16,17]. However, the composition of smoke extracts from these resins remains largely unexplored.

Frankincense and myrrh, as well as the bark of *B. papyrifera*, are primarily used in traditional Bakhoor therapy in Sudanese. In addition, some individuals soak frankincense in water and drink the solution as an expectorant. While nourishing the skin, water extracts of both frankincense and myrrh are applied to the skin to treat conditions like acne, paleness, and redness. Additionally, frankincense and myrrh are chewed, with the bitter compounds believed to help alleviate dental and periodontal issues, and swallowing the resin is thought to offer health benefits [5,9]. Both *Boswellia* and *Commiphora* resins have a long history of use in traditional medicine across several countries and regions, including India, China, Rome, and Greece [18]. In addition, resins from these species, along with the bark of Boswellia spp., are well regarded in traditional Sudanese medicine and inhalation therapies (Figure 1A) for treating oral diseases, skin whitening, and antibacterial properties. However, the scientific validation of inhalation and smoke therapy for addressing health conditions prevalent in traditional Sudanese medicine is still lacking.

On the basis of these background, this study focused on two objectives. First, we analyzed aromatic compounds from smoke using various extraction methods. Given the limited research on solid-phase microextraction and smoke-trapping techniques, we aimed to identify and compare the volatile compounds present in these substances. The parallel utilization of different extraction methods led us to more comprehensive understanding of the volatile compounds generated from the resin and bark of *B. papyrifera* and the resin of *C. africana*. Second, we evaluated their biological activities to support their potential as traditional therapeutic and aesthetic agents (Figure 1B). We focused on biological activities relevant to dermal, oral, and abdominal disorders. Tyrosinase is the key enzyme for the melanin formation in melanocyte in the basal cell layer of human epidermis. The chemicals in the smoke extracts can directly contact with the melanin synthesis system in the epidermis, and both frankincense and myrrh are directly applied to the skin for nourishing. Thus, we included inhibition of tyrosinase in the bioassays, based on the strong attention to skin whitening. In addition, given that cancers account for a significant proportion of human mortality, we investigated the antiproliferative activity of these materials against cancer cells to evaluate their potential contribution to health in association with dermal and abdominal issues. We targeted human melanoma cells (G361), lung cancer cells (A549), and pancreatic cancer cells (MIA Paca-2). Further, the growth inhibition of *Streptococcus mutans* Clarke, a bacterium linked to oral disease [19,20], was also included in the bioassays. Since the resins are used as chewing gum, the chemicals in the resins can directly affect microorganisms in the mouth. To the best of our knowledge, this is the first study to explore the biological activities of smoke extracts derived from the resin and bark of *Boswellia papyrifera* and the resin of *Commiphora africana*, offering foundational insights into their potential health benefits.

## 2. Results

### 2.1. GC-MS Detection of the Volatiles

Samples were prepared using three extraction methods from the three materials and expressed as a combination of capital letters (**M**, **K**, and **S**) and numbers (**1**–**3**). Capital letters indicate the different extraction methods: MonoTrap (**M**), smoke (**K**), and solvent (**S**) extractions, whereas numbers indicate the differences in materials: *C. africana* resin (**1**), *B. papyrifera* resin (**2**), and *B. papyrifera* bark (**3**).

The samples were subjected to GC-MS analysis, and peaks with a peak area greater than 0.5% of the total area of detected peaks were picked up (Table 1, Table 2 and Table 3). Numerous volatile compounds were detected in the samples of *C. africana* and *B. papyrifera*, and the chromatograms demonstrated differences in the volatile compounds between *C. africana* and *B. papyrifera* resin samples (Figure 2 and Figure 3). The resin and bark samples of *B. papyrifera* had similar volatile compound compositions (Figure 3 and Figure 4). Moreover, differences in the extraction methods caused large variations in the chromatographic profiles. The chromatograms of the MonoTrap extracts showed compounds with short retention times. In contrast, chromatograms of smoke extracts showed the presence of compounds with moderate retention times in the GC-MS analysis, and solvent extracts contained compounds with a wide range of retention times.

Volatile compounds were identified by comparing the mass spectra of the detected peaks with those in the DIST08 mass spectral library database, which resulted in the detection of a number of known and unknown compounds (Table 1, Table 2 and Table 3). The identities of compounds, sabinene, limonene, *p*-cymene, bornyl acetate, verbenone, octyl acetate, (*E*)-nerolidol, and β-eudesmol were confirmed by the comparison of retention time with the authentic compounds (Figure 5).

### 2.2. Compounds Detected from C. africana Resin, B. papyrifera Resin, and B. papyrifera Bark Samples

The results of analyses of extracts from *C. africana* resin are shown in Figure 2 and Table 1. The major compound in **M1** was limonene with a peak area of 24.8%, followed by sabinene (19.9%). Verbenone (19.2%) and β-selinene (18.0%) showed the topmost peaks in **K1**. Verbenone (14.9%) and β-selinene (8.3%) were also the major peaks in **S1**.

The compounds detected in *B. papyrifera* resin are listed in Table 2. Octyl acetate and limonene were the major compounds in **M2**, with peak areas of 60.0% and 14.5%, respectively. In **K2**, the largest peaks were nerolidol isobutyrate with a peak area of 35.7% and 12-(acetoxy)-2,6,10-trimethyl-(*E*,*E*,*E*)-2,6,10-dodecatrien-1-al with a peak area of 20.1%. The major peaks in **S2** were nerolidol isobutyrate and 12-(acetoxy)-2,6,10-trimethyl-(*E*,*E*,*E*)-2,6,10-dodecatrien-1-al with respective peak areas of 11.2% and 7.8%. Figure 3 shows the chromatograms of the *B. papyrifera* resin from MonoTrap, smoke, and solvent extracts. Octyl acetate and *trans*-nerolidol were detected in **M2**, **K2**, and **S2**, whereas limonene was detected only in **M2**.

The chromatograms of the *B. papyrifera* bark extracts **M3**, **K3**, and **S3** (Figure 4) showed a high similarity to those of the resin extracts. As listed in Table 3, major peaks in **M3** were octyl acetate (72.9%) and *trans*-nerolidol (8.2%), whereas major peaks in **K3** were 12-(acetoxy)-2,6,10-trimethyl-(*E*,*E*,*E*)-2,6,10-dodecatrien-1-al (27.5%), nerolidol isobutyrate (26.7%), and *trans*-nerolidol (13.4%). The solvent extract (**S3**) contained 12-(acetoxy)-2,6,10-trimethyl-(*E*,*E*,*E*)-2,6,10-dodecatrien-1-al (9.8%), and nerolidol isobutyrate (7.3%) as major compounds.

The amounts of volatile compounds absorbed by a MonoTrap disk are small and not sufficient for bioassays. Thus, the smoke (**K1**–**K3**) and solvent (**S1**–**S3**) extracts of *C. africana* resin, *B. papyrifera* resin, and *B. papyrifera* bark were subjected to bioassays to detect their biological activities.

### 2.3. Tyrosinase Inhibition

The tyrosinase activity was determined in the presence of the smoke extracts, **K1**, **K2**, and **K3**, to examine their inhibitory activity (Figure 6). The reaction with high concentrations of the extracts resulted in the appearance of turbidity of the reaction mixture. Thus, the maximum concentration was set at 10% for accurate evaluation of enzyme activity by avoiding the turbidity generation. The inhibition rates of **K1** were 66.7% at 10% and 39.1% at 2.5%. On the other hand, the inhibition rates of **K2** and **K3** were 7.5% and 8.2%, respectively, at 10%. Thus, **K1** exhibited the highest tyrosinase inhibitory activity among the smoke extracts tested. We also attempted to perform a bioassay with solvent extracts **S1**, **S2**, and **S3**; however, the turbidity of the reaction mixture appearing after adding these extracts prevented us from determining tyrosinase activity.

Next, we examined the tyrosinase inhibitory activities of commercially available authenticated compounds *p*-cymene, (*S*)-limonene, (*R*)-limonene, verbenone, octyl acetate, and *E*-nerolidol (Figure 7A). The concentration range for each compound in this experiment was determined to avoid the appearance of colored materials generated by oxidation of the compound, considering the limitations of compound availability. *p*-cymene exhibited the highest tyrosinase inhibitory activity among the tested compounds. (*S*)-limonene also exhibited inhibitory activity at high concentrations. In addition, we examined the inhibitory activities of the structural isomers of *p*-cymene including *o*-cymene, *m*-cymene, and cumene, which lacks the methyl group on the benzene ring of cymenes. *p*-cymene and *m*-cymene displayed similar inhibition rates, whereas *o*-cymene and cumene had lower inhibition rates (Figure 7B).

We analyzed the mode of inhibition of *p*-cymene on the diphenolase activity of tyrosinase using L-DOPA as the substrate. The Dixon plot with *p*-cymene showed the intersection of the regression lines with and without the inhibitor in the second quadrant, indicating that the mode of inhibition was competitive inhibition (Figure 7C). Similarly, the inhibition mode of (*S*)-limonene was competitive inhibition (Figure 7D). The *K*_i_ values for *p*-cymene and (*S*)-limonene were 4.10 and 6.50 mM, respectively.

### 2.4. Effects on the Cell Viability of Cancerous and Normal Cell Lines

We treated human melanoma cells G361, lung cancer cells A549, pancreatic cancer cells MIA Paca-2, and normal human skin fibroblasts NB1RGB with smoke extracts of *C. africana* resin (**K1**), *B. papyrifera* resin (**K2**), and *B. papyrifera* bark (**K3**). The treatment with **K1** at 0.1% showed the inhibitory activity in A549 and MIA Paca-2 cells (Figure 8A). In particular, MIA Paca-2 cells were sensitive to the treatment with a 60% inhibition observed. **K2** and **K3** failed to suppress the cell growth even at 0.1% in all cells. The treatments with smoke extracts at 0.03% did not affect the cell viability on all cell lines.

The treatment with solvent extracts of *C. africana* resin (**S1**) and *B. papyrifera* resin (**S2**) at 0.1% showed the strong inhibition of cell viability as compared to that of smoke extracts with the almost complete inhibition of all cell lines (Figure 8B). The treatment of **S1** at 0.03% suppressed the cell growth of A549 and MIA Paca-2 by 25% and 37%, respectively. The treatment with **S2** at 0.03% tended to inhibit the growth of cancerous cell lines, G361, A549, and MIA Paca-2, but not fibroblast cells. **S3** slightly suppressed the cell growth of G361, A549, and MIA Paca-2 only at 0.1%.

### 2.5. Effects on the S. mutans

We confirmed beforehand that the culturing *S. mutans* cells in BHI medium containing 1% or 5% DMSO had no significant effect on the growth. Thus, we examined the effects of extracts with the DMSO concentrations not exceeding 5%. The treatments of the bacteria with smoke extracts from *C. africana* resin (**K1**) and *B. papyrifera* resin (**K2**) and bark (**K3**) at concentrations lower than 1% did not suppress the bacterial growth (Figure 9A). On the other hand, the treatments with **S1** and **S2** at 5% markedly inhibited the bacterial growth compared to the control treatment without these extracts (Figure 9B). Unexpectedly, the treatment with solvent extracts of *C. africana* resin (**S1**), *B. papyrifera* resin (**S2**), and bark (**S3**) at 1% significantly promoted the growth of *S. mutans*.

## 3. Discussion

This study aimed to characterize the volatile compounds in the resin of *C. africana* and *B. papyrifera*, in addition to those in the bark of *B. papyrifera*, through the MonoTrap solid-phase microextraction and smoke trapping techniques, as well as acetone immersion. The MonoTrap technique was reported to rapidly extract polar or nonpolar compounds under high or low boiling points [21]. In contrast, smoke trapping techniques expose the samples to high temperatures for a long time, while acetone immersion extracts the volatiles at room temperature. The difference of extraction methods was clearly reflected by chromatograms obtained by GC-MS analyses. The MonoTrap samples (**M1**–**M3**) efficiently retained low-boiling-temperature compounds that were eluted before 10 min, whereas the smoke trapping (**K1**–**K3**) captured high-boiling-temperature compounds that were eluted after 14 min. The samples from solvent immersion (**S1**–**S3**) contained compounds that were eluted after a retention time of 19 min, which are not likely present in the smoke from these materials. These findings indicated that three extraction methods are complimentary to each other.

The GC-MS analysis of the extracts prepared from *C. africana* and *B. papyrifera* revealed that the constituents of both species were markedly different, although resins from these plants were used in similar ways. The resin and bark of *B. papyrifera* showed relatively simple chromatograms, with limonene, octyl acetate, and trans-nerolidol being the main fragrant compounds. In contrast, the chromatograms of *C. africana* showed many peaks, with verbenone, sabinene, β-selinene, limonene, and *cis*-verbenol being detected. Limonene, β-eudesmol, acetic acid, and nerolidol isobutyrate were commonly detected in *C. africana* gum-resin and *B. papyrifera* gum-resin and bark. The presence of these common fragrant constituents underpins the widespread use of these materials. Delving into the nuanced distinctions in the utilization contexts of *C. africana* and *B. papyrifera* presents an intriguing issue for further investigation.

The detected compounds in this study agreed with the findings of DeCarlo et al. [15] and Dinku et al. [16], who detected several monoterpenes, diterpenes, and sesquiterpenes, including *p*-cymene, limonene, bornyl acetate, verbenone, copaene, caryophyllene, and β-selinene in *C. africana* and *B. papyrifera* samples by GC-MS analysis coupled with hydro-distillation. The present study detected additional compounds, such as tetrachlorohydroquinone dimethyl ether in *C. africana*, 12-(acetoxy)-2,6,10-trimethyl-(*E*,*E*,*E*)-2,6,10-dodecatrien-1-al and 4-methylene-1-methyl-2-(2-methyl-1-propen-1-yl)-1-vinylcycloheptane in *B. papyrifera*, and nerolidol isobutyrate in both species. The difference in the volatile compounds between the present and previous researches might result from various factors, such as the tree’s origin, environmental conditions, geographic differences, and season of collection [22].

Water extracts of *C. africana* and *B. papyrifera* resins have been used for nursing the skin in Sudanese culture. In this context, we analyzed the effect of the extracts on tyrosinase, which is involved in melanin formation in basal cells of the epidermal tissue. We found strong inhibitory activity against tyrosinase in **K1**, a smoke extract of *C. africana* resin. Although antioxidant and antimicrobial activities have been previously reported [23], this is the first report of the antityrosinase activity of *C. africana* resin extract.

Among the compounds tested, *p*-cymene and (*S*)-limonene were the most effective inhibitors. We also found that *p*-cymene and *m*-cymene exhibited higher inhibition rates than *o*-cymene and cumene. The modes of tyrosinase inhibition by *p*-cymene and (*S*)-limonene were competitive inhibition. α-Arbutin and 4-hydroxyphenylacetic acid are well-known competitive inhibitors of tyrosinase. Their inhibitory activity is attributed to the presence of phenolic groups on the benzene ring that are also present in the substrate L-DOPA [24]. Interestingly, *p*-cymene and (*S*)-limonene, which do not have phenolic groups and are distinct in terms of chemical structure, inhibited tyrosinase through competitive inhibition. The smoke extract **K1** did not contain significant amounts of *p*-cymene and (*S*)-limonene, indicating the presence of additional inhibitors in the extract.

The extracts of *C. africana* (**S1**) and *B. papyrifera* (**S2**) resins showed potent antiproliferative effects on cell lines, including the human melanoma cell line G361, lung cancer cell line A549, pancreatic cancer cell line MIA Paca-2, and normal human skin fibroblasts NB1RGB at a 0.1% concentration. The smoke extract **K1** also suppressed cell growth at a 0.1% concentration to some extent, whereas **K2** and **K3** failed to suppress the cell growth. It is of interest to note that MIA Paca-2 was more sensitive to **K1** as compared to A549 and G361. This might be due to the expression of mutant p53 in MIA Paca-2 [25], instead of wild type p53 in A549 and G361 [26,27], because some natural products including triptolide and yunnanterpene D have been shown to exert anticancer activity by targeting mutant p53 [28]. Furthermore, we found the presence of sabinene, limonene, eucalyptol, and *p*-cymene in the extract of *C. africana* resin (**S1**), but not in the smoke extract (**K1**). The strong inhibitory activity observed in **S1** may be attributable to these compounds because they have been shown to have antiproliferative activity of various cell lines [29,30,31]. Mwangi et al. examined the cytotoxicity of hexane, dichloromethane, ethyl acetate, and methanol extracts of *C. africana* bark and found that the hexane extract inhibited the growth of Vero cells [32].

Among the compounds detected in the extracts, the mode of action of limonene and β-elemene have been investigated extensively. D-limonene demonstrated to have anti-proliferative activity against many types of cancer cells [33], and the activity has been implicated in enhancement of the pathway leading to apoptosis. For instance, the treatment of transplanted lung tumors with D-limonene induced the expression of apoptosis- and autophagy-related genes along with the suppressed growth of tumors in nude mice [34]. In addition, apoptosis has been identified to be most possible reason behind anticancer activity exerted by β-elemene against a variety of cancer cell lines [35]. β-elemene has also been shown to arrest the cell cycle at G2/M in human NSCLC cell lines (H460 and A549) [36]. The biochemical analysis of the pathways activated in cancer cells treated with the extracts from *C. africana* and *B. papyrifera* will provide further insights into the functional pathway leading to the cell death and responsible molecules in the extract.

The composition of *B. papyrifera* resin extract (**S2**) was almost similar to those of the smoke extract (**K2**) and bark extract (**S3** and **K3**) in the chromatograms obtained by GC-MS analyses. The difference in the inhibitory effect on cell growth may be ascribable to differences in the concentrations of compounds in the extracts, although the possibility of the involvement of compounds that were not detected by GC-MS was not excluded. Boswellic acids are pentacyclic triterpenes that are characteristic of the genus *Boswella* including *B. papyrifera* [37] and have attracted significant attention because of their anticancer properties [38]. Our GC-MS analysis was not suitable for the detection of boswellic acids because of their large molecular weights. To address the chemical basis of the inhibitory activity, the identification of active compounds in the extract is needed for *B. papyrifera* extracts (**S2**, **S3**, **K2**, and **K3**).

In addition to their application in inhalation therapy, the resins of *C. africana* and *B. papyrifera* are consumed as chewing gum in African countries. The Gram-positive bacterium *S. mutans* is the predominant microorganism involved in the initiation of dental caries [20] and plays a major role in tooth decay and the metabolism of sucrose to lactic acid. Solvent extracts of **S1** and **S2** promoted *S. mutans* growth at a 1% concentration, whereas they significantly inhibited *S. mutans* growth at 5%. Thus, these solvent extracts may contain both substances that promote and inhibit the growth of *S. mutans*, although no substances that promote the growth of *S. mutans* has been reported heretofore. Among the compounds detected in the extracts, β-caryophyllene, verbenone, β-myrcene, β-caryophyllene, sabinene, linalool, and nerolidol have been indicated to have inhibitory activity on the growth of S*. mutans* [39,40,41,42]. Furthermore, terpenoids, such as limonene, eucalyptol, bornyl acetate, β-elemene, verbenone, *cis*-verbenol, and caryphyllene oxide in **S1** and β-eudesmol contained in **S1** and **S2**, have been shown to exhibit inhibitory activity against various bacteria [43,44,45,46,47,48,49,50]. Terpenoids are a large group of phytochemicals with promising functional pharmaceutical activities [16,51]. Several recent reports mentioned that 75% of antibacterial drugs are terpenes, including classes of mono-, di-, tri-, and sesquiterpenoids [52]. The lipophilicity or hydrophobicity of terpenes and their carbonyl and hydroxyl groups are among the determining factors of their various pharmaceutical actions [53]. It is highly likely that terpenoids specifically detected in **S1** and **S2** contribute to the inhibition of *S. mutans* growth by *C. africana* and *B. papyrifera* extracts. Furthermore, the GC-MS chromatogram revealed that solvent extracts (**S1**–**S3**) contained compounds that were not detected in the MonoTrap (**M1**–**M3**) or smoke (**K1**–**K3**) samples. Consequently, the potent inhibitory activities observed in **S1** and **S2** could be linked to the presence of those compounds specifically detected in solvent extracts. Nonetheless, arduous analyses are essential to understand the holistic biological impacts of the constituents of *C. africana* and *B. papyrifera* materials and their mechanisms.

## 4. Materials and Methods

### 4.1. Sample Preparation

The resin of *Commiphora africana* (A.Rich.) Endl. and the resin and bark of *Boswellia papyrifera* (Delile ex Caill.) Hochst. were obtained from herb dealers in the Omdurman local market in Sudan. The sample was classified and authenticated at the Herbarium of Medicinal and Aromatic Plants and Traditional Medicine Research Institute—the National Centre for Research Sudan by the taxonomist Yahya Sulieman Mohamed Sulieman. The samples were then cleaned, washed, shade-dried and powdered. Each sample was stored in a plastic bag at 25 °C until further analysis.

### 4.2. Collection of Volatiles and Aromatic Compounds

Volatile compounds were collected using three methods: (1) MonoTrap solid-phase microextraction of volatiles generated at low temperatures; (2) trapping smoke from burning materials with acetone by mimicking traditional Sudanese usage; and (3) solvent extraction with acetone (Figure 1B).

MonoTrap extraction of volatile compounds was performed using a MonoTrap kit, according to the manufacturer’s guidelines (Monolithic Silica Adsorbents, Merck KGaA, Darmstadt, Germany). Four grams of samples was weighed and added to MonoTrap stand vials. An RCC18 rod MonoTrap (5 mm × 2.9 mm thickness, Merck KGaA) was inserted into the attached holders using clean tweezers, and then, the holders were inserted into MonoTrap stand vials through the vial cap for volatile adsorption. The vials were then placed in a water bath at 70 °C for 40 min. Subsequently, the RCC18 rod MonoTrap was placed into a MonoTrap extraction cup filled with an extraction solvent (200 μL acetone). Subsequently, the septum was tightened, and the vials were placed in an ultrasonicator for 15 min at 20 °C. One microliter of the solution was subjected to gas chromatography–mass spectrometry (GC-MS) analysis. The extracted samples were named **M1** (MonoTrap, *C. africana* resin), **M2** (MonoTrap, *B. papyrifera* resin), and **M3** (MonoTrap, *B. papyrifera* bark). All samples were stored at 4 °C for further analyses.

Smoke trapping was performed in accordance with the method described by Eltigani et al. [54], with slight modifications. Four grams of the sample was placed at the bottom of a double-necked round-bottom flask (sample flask). A valve for the airflow control was connected to the left side of the flask. A trapping solvent (10 mL acetone) was added to a conical flask (trapping flask) capped with a Teflon cap perforated with two glass tubes. The end of one glass tube was placed below the solvent surface, and the end of the other tube was placed in the headspace. The right side of the sample flask was connected to a glass tube using a Teflon tube, the end of which was placed below the solvent in the trapping flask. Another glass tube attached to the trapping flask was connected to an aspirator. The round-bottom flask was heated using a Bunsen flame, and the smoke produced in the round-bottom flask was introduced into the trapping solvent using an aspirator. Heating was stopped before the total burning of the samples.

The trapping solvent was then dried overnight in Na_2_SO_4_ at room temperature. The acetone solution was filtered. A control sample was prepared using acetone without adding the wood sample to a double-necked round-bottom flask to detect volatiles from the glass and plastic ware. The samples were labeled **K1** (smoke, *C. africana* resin), **K2** (smoke, *B. papyrifera* resin), and **K3** (smoke, *B. papyrifera* bark). The acetone solution containing volatiles (10 mL) from the samples was concentrated to 1 mL using nitrogen gas. All samples were stored at 4 °C for further analyses. For the analysis of the biological activities, DMSO (0.5 mL) was added to an acetone solution (5 mL) containing the volatiles. After the removal of acetone by evaporation, the remaining DMSO solution was used for the bioassays.

Four grams of resin and bark samples was soaked in 10 mL of acetone for 30 min at room temperature with slow shaking. The mixtures were filtered and labeled **S1** (solvent extraction, *C. africana* resin), **S2** (solvent extraction, *B. papyrifera* resin), and **S3** (solvent extraction, *B. papyrifera* bark). Stock solutions were prepared by dilution of the extracts with acetone to 10 folds and used for GC-MS analysis. For the analysis of biological activities, 1 g of the samples was extracted from 20 mL acetone in the same way. After adding DMSO (1 mL) to the solution, acetone was removed by evaporation.

### 4.3. GC-MS Analysis

GC-MS analysis (Shimadzu GCMSQP2010C, Kyoto, Japan) was performed using an Agilent J&W DB-WAX column (thickness, 0.25 μm; length, 30 m; inner diameter, 0.25 mm; Agilent, Santa Clara, CA, USA). The initial oven temperature was 45 °C, and the mass range was *m/z* 50–600. The injection volume was 1 μL, and the injector port temperature was 280 °C. The following oven temperature program was applied using helium (at a constant flow rate of 1 mL min^−1^) as the carrier gas: 5 min at 45 °C, increased to 280 °C at 12 °C min^−1^, and then to 220 °C at 4 °C min^−1^. The interface and ion source were maintained at 220 and 200 °C, respectively. All mass spectra were acquired in the electron impact mode. Ionization was turned off during the first 5 min to avoid solvent overloading.

The mass spectra of the detected peaks were compared to those in the DIST08 database. The predicted compounds with similarity indices less than 80% were reported as unknown compounds. The identities of some compounds were confirmed by comparing their retention times in the GC-MS analysis with those of authentic compounds. The analyzed compounds were as follows: *p*-cymene, β-eudesmol (*S*)-limonen, (*R*)-limonen, and octyl actetate obtained from Fujifilm Wako (Osaka, Japan), bornyl acetate and verbenone from TCI (Tokyo, Japan), and sabinen and (*E*)-3,7,11-trimethyldodeca-1,6-10-trien-3-ol from BLDpharm (Shanghai, China).

### 4.4. Tyrosinase Inhibitory Activity

The sample solution (20 µL, DMSO for the blank) was added to a mixture of water (30 µL), 0.1 M HEPES buffer (pH 7.0, 50 µL), and a tyrosinase solution (100 U/mL, 50 µL; from mushroom, Sigma-Aldrich) in a well of a 96-well microplate and mixed well. After adding a 2.5 mM solution (50 µL) of L-DOPA (Nacalai Tesque, Kyoto, Japan), the absorbance at 475 nm was measured immediately (0 min) and after 5 min using an Infinite 200 PRO microplate reader (Tecan Japan Inc., Tokyo, Japan). The tyrosinase inhibition rate was calculated using the following equation:Inhibition rate (%) = (1 − A/B) × 100(1)
where A is the difference between the absorbance of the reaction solution with the inhibitor immediately after the addition of L-dopa and after 5 min and B is the difference between the absorbance of the reaction solution without the inhibitor immediately after the addition of L-DOPA and after 5 min.

### 4.5. Cell Culture

The human melanoma cell line G361, human lung cancer cell line A549, pancreatic cancer cell line MIA Paca-2, and normal human skin fibroblast line NB1RGB were obtained from Riken Bioresource Center (Tsukuba, Japan). The cells were cultured in Dulbecco’s modified Eagle’s medium-high glucose (DMEM-high glucose, Fujifilm Wako) supplemented with 10% bovine fatal serum, 100 U/mL streptomycin (Meiji Seika, Tokyo, Japan), and 100 μg/mL penicillin (Meiji Seika) in a CO_2_ incubator under an atmosphere of 95% air with 5% CO_2_.

### 4.6. Detection of Cell Viability Using the WST-8 Assay

Cell viability was assessed using the Cell Counting Kit-8 (WST-8; Dojindo, Kumamoto, Japan), in accordance with the method by Tanaka et al. [55]. Briefly, the cells (5 × 10^3^ cells/well) were seeded in a 96-well culture plate on the day before the experiment. After the administration of each extract at final concentrations of 0.3% and 0.1%, the cells were incubated at 37 °C for 48 h. Cell viability was evaluated by measuring the absorbance with a microplate spectrometer (Sunrise^TM^ Fuji film Wako Pure Chemicals, Osaka, Japan) at 450 nm between 1 and 4 h after the addition of the WST-8 reagent to the culture medium in each well. The ratio of each dataset to the control was calculated.

### 4.7. Evaluation of the Antibacterial Activity Against S. mutans

*S. mutans* MT8148 was obtained from RIKEN BRC (Tsukuba, Japan) and grown in a brain heart infusion (BHI) broth (Shimadzu Diagnostics Corporation, Tokyo, Japan). Fifty microliters of *S. mutans* glycerol stock was inoculated into 5 mL of BHI medium and incubated statically at 37 °C for 24 h. Thereafter, a pre-cultured *S. mutans* suspension (10 µL) was added into microtiter plate wells that contained 50 µL of 2× BHI medium and 40 µL of samples diluted with sterile distilled water, and the mixture was statically cultured at 37 °C for 24 h. The final concentration of DMSO in the medium was prepared to be 1% or 5%.

Bacterial growth was evaluated using a BacTiter-Glo^TM^ Microbial Cell Viability Assay kit (Promega, Madison, WI, USA) in accordance with the manufacturer’s instructions. Chemiluminescence was measured using a luminometer (AB-2270 Luminescencer Octa, ATTO Corp, Tokyo, Japan).

### 4.8. Statistical Analyses

The data obtained for the tyrosinase inhibitory activities were analyzed by Tukey−Kramer test, while those against the viability of cancerous cells were analyzed by Dunnett T3 test. The inhibitory activities on bacterial growth were analyzed by the two-tailed Student’s *t* test. The analyses were performed on BellCurve for Excel (Social Survey Research Information Co., Ltd., Tokyo, Japan), SPSS Statistics software ver. 29 (IBM Japan, Tokyo, Japan), and Microsoft Excell (Redmond, WA, USA). All data were expressed as mean ± standard deviation, and the differences were considered statistically significant at *p* < 0.05.

## 5. Conclusions

We conducted a comprehensive and comparative analysis of extracts from *C. africana* and *B. papyrifera*, employing three different extraction techniques. The analyses revealed that utilization of multiple extraction methods was effective for understanding the composition of the volatile compounds in these materials. The compounds were identified using GC-MS, revealing shared and distinctive volatile compounds in the extracts.

Based on the traditional use of the resins and bark in Sudanese inhalation therapy to treat diseases and to enhance skin health, as well as in the practice of chewing the resins for oral hygiene and overall health, we investigated a range of biological activities. The examined activities included inhibition of tyrosinase, cytotoxicity against cancer cell lines, and antimicrobial effects against *S. mutans*. Our results from exploring tyrosinase inhibition highlighted specific monoterpenoids, such as *p*-cymene and (*S*)-limonene, which act as competitive inhibitors. Additionally, the extracts **S1** and **S2** exhibited the most potent inhibition on cancer cell viability and bacterial survival, indicating involvement of compounds with low volatility.

This study underscores the potential for discovering novel biological activities and compounds through rigorous analysis of traditional materials used as medicines. The diverse range of activities observed highlights the promising applications of *C. africana* and *B. papyrifera* resins and barks, supporting the validity of their traditional uses. Our findings enhance scientific understanding of these natural resources and suggest opportunities for further research and application in fields such as medicine and industry.

## Figures and Tables

**Figure 1 molecules-30-00499-f001:**
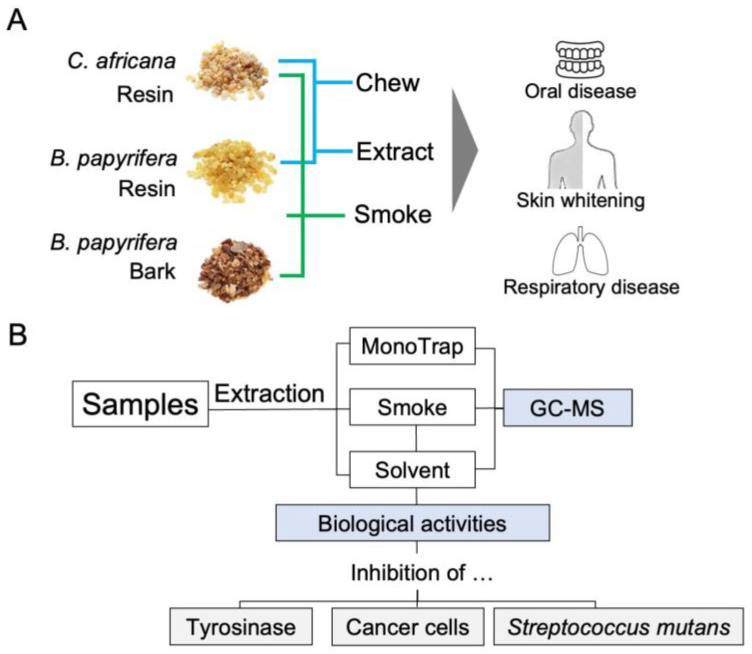
Schematic representation of the utilization of resin and bark in traditional Sudanese medicine (**A**) and the extract preparation and their analyses (**B**).

**Figure 2 molecules-30-00499-f002:**
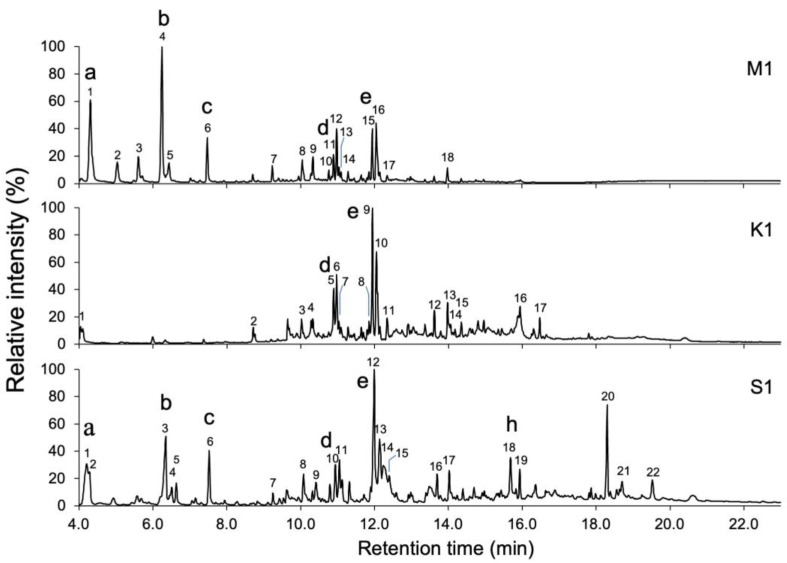
GC-MS analyses of the extract of resin of *Commiphora africana*. Extracts were prepared by MonoTrap solid-phase microextraction (**M1**), by trapping smoke generated from burned resin (**K1**), and by solvent extraction with acetone (**S1**). The numbers above the peaks correspond to the compounds listed in Table 1, Table 2 and Table 3, and the lowercase letters above the peaks indicate the compounds depicted in Figure 5.

**Figure 3 molecules-30-00499-f003:**
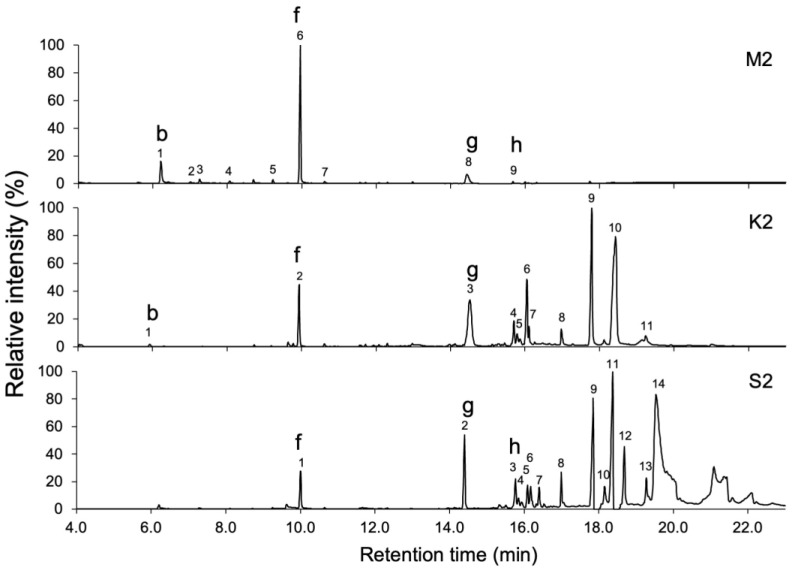
GC-MS analyses of the extract of resin of *Boswellia papyrifera*. Extracts were prepared by MonoTrap solid-phase microextraction (**M2**), by trapping smoke generated from burned resin (**K2**), and by solvent extraction with acetone (**S2**). The numbers above the peaks correspond to the compounds listed in Table 1, Table 2 and Table 3, and the lowercase letters above the peaks indicate the compounds depicted in Figure 5.

**Figure 4 molecules-30-00499-f004:**
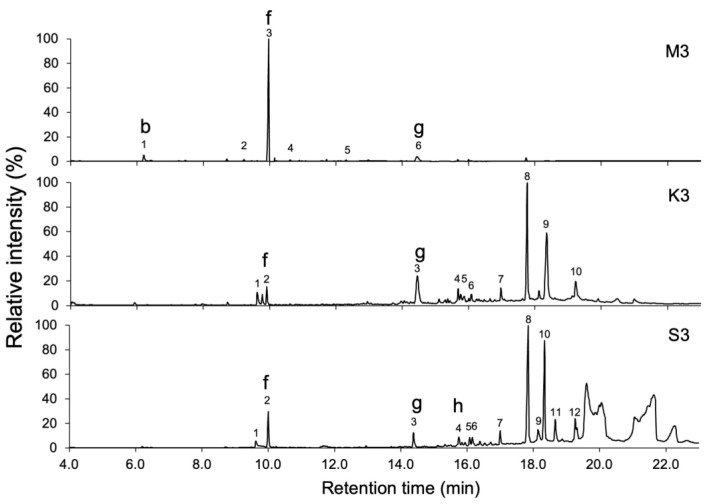
GC-MS analyses of the extract of bark of *Boswellia papyrifera*. Extracts were prepared by MonoTrap solid-phase microextraction (**M3**), by trapping smoke generated from burned resin (**K3**), and by solvent extraction with acetone (**S3**). The numbers above the peaks correspond to the compounds listed in Table 1, Table 2 and Table 3, and the lowercase letters aboev the peaks indicate the compounds depicted in Figure 5.

**Figure 5 molecules-30-00499-f005:**
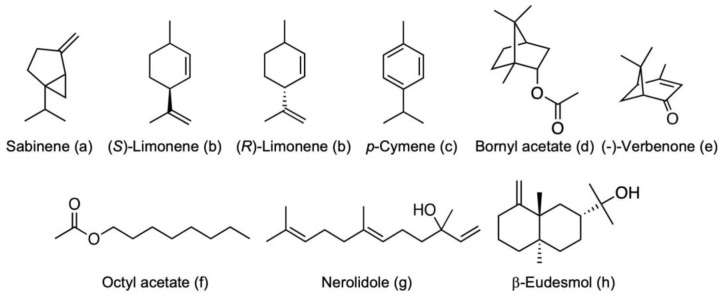
Compounds detected in the extracts of resin of *Commiphora africana* and resin and bark of *Boswellia papyrifera.* These compounds were used for the tyrosinase inhibition assay.

**Figure 6 molecules-30-00499-f006:**
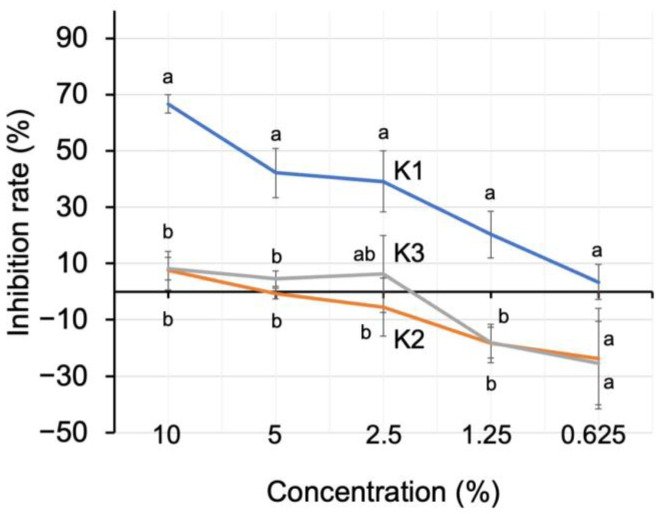
Effects of the smoke extracts from the resin of *Commiphora africana* (**K1**) and from the resin (**K2**) and bark (**K3**) of *Boswellia papyrifera* on tyrosinase activity. Data in graphs are presented as mean ± SD from five replicates. Different letters indicate statistical differences among the samples from the same concentrations (*p* < 0.05; Tukey−Kramer test).

**Figure 7 molecules-30-00499-f007:**
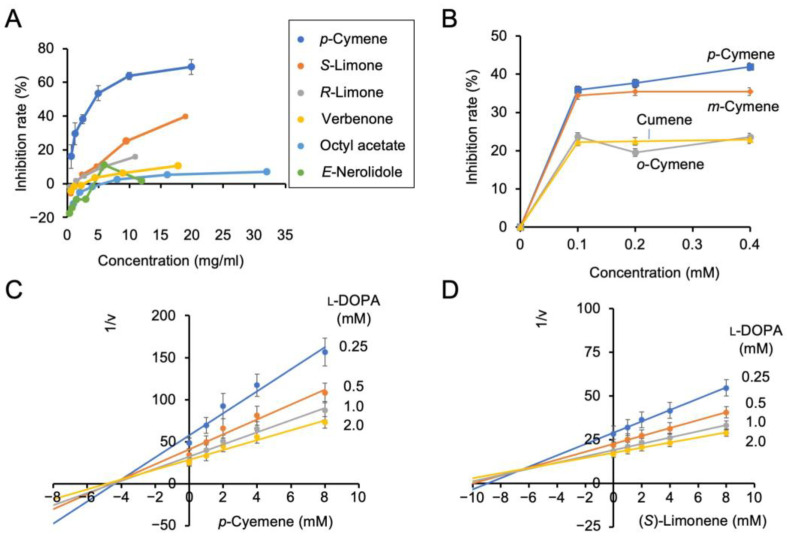
Inhibitory activity of *p*-, *m*-, and *o*-cymene and cumene on tyrosinase: (**A**) effects of the compounds detected in the extracts on tyrosinase activity; (**B**) inhibition of tyrosinase activity by *p*-, *m*-, and *o*-cymene and cumene at various concentrations; (**C**) Dixon plot of inhibition of tyrosinase by *p*-cymene at 1, 2, 4, and 8 mM; (**D**) Dixon plot of inhibition of tyrosinase by (*S*)-limonene at 1, 2, 4, and 8 mM. Data in graphs are presented as mean ± SD from three replicates.

**Figure 8 molecules-30-00499-f008:**
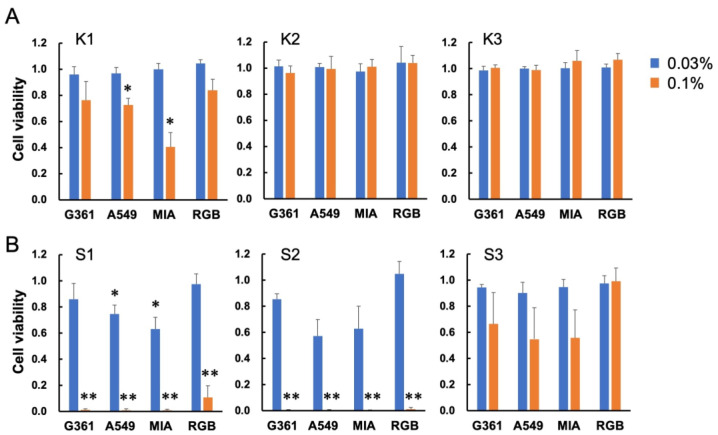
Effects of the smoke extracts (**A**) and solvent extracts (**B**) from the resin of *Commiphora africana* and from resin and bark of *Boswellia papyrifera* on cancerous cell lines. Smoke extracts (**K1**, **K2**, and **K3**) and solvent extracts (**S1**, **S2**, and **S3**) were added to the culture at 0.03% and 0.1%. The cell viability was determined 48 h after adding the extract by the WST-8 method. Data in graphs are presented as mean ± SD from three replicates. Asterisks indicate a statistical difference from the control. (* *p* < 0.05, ** *p* < 0.01; ANOVA followed by Dunnett T3 test).

**Figure 9 molecules-30-00499-f009:**
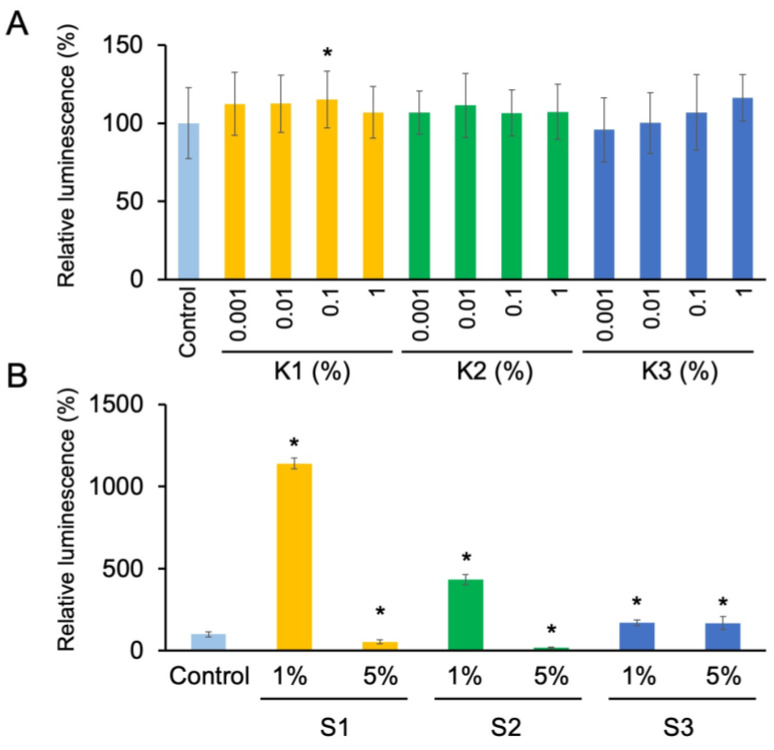
Effects of the extracts from the resin of *Commiphora africana* and from the resin and bark of *Boswellia papyrifera* on *Streptococcus mutans* growth: (**A**) smoke extracts (**K1**, **K2**, and **K3**) were added to the media. DMSO (1% at final concentration) served as the control; (**B**) solvent extracts (**S1**, **S2**, and **S3**) were added to the media. In control experiments, extracts were not added to the media. Bacterial viability was evaluated 24 h after adding the extracts. Data in graphs are presented as mean ± SD from at least eight replicates. Asterisks indicate a significant statistical difference from the control (* *p* < 0.05; Student’s *t* test).

**Table 1 molecules-30-00499-t001:** Compounds detected in *C. africana* resin extracts.

No.	Compound	R.T. (min)	Area (%)	Major Ions (*m/z*)	Similarity Index
	**M1**				
1	Sabinene (a)	4.3	19.9	41, 77, 93, 136	95
2	3-Carene	5.0	4.0	43, 79,93, 136	95
3	β-Myrcene	5.6	4.0	41,53, 69, 93	96
4	Limonene (b)	6.2	24.8	68, 93, 107, 121	94
5	Eucalyptol	6.4	3.9	43, 81, 108, 154	91
6	*p*-Cymene (c)	7.5	5.7	65, 91, 119, 134	94
7	1-Methoxy-1,3,5-cycloheptatriene	9.2	1.7	77, 91, 107, 122	90
8	6-Camphenol	10.0	3.0	67, 93, 108, 119	91
9	α-Bourbonene	10.3	2.6	81, 105, 123, 161	92
10	Pinocarvone	10.8	0.9	50, 81, 108, 150	89
11	Bornyl acetate (d)	10.9	3.2	43, 95, 121, 136	96
12	β-Elemene	11.0	5.7	81, 147, 189, 204	94
13	Caryophyllene	11.0	1.6	91, 119, 133, 175	93
14	5-Isopropyl-2-methylbicyclo[3.1.0]hexan-2-ol	11.1	0.9	71, 91, 111, 154	84
15	Verbenone (e)	11.9	5.3	107, 122, 135, 150	95
16	β-Selinene	12.1	8.8	105, 175, 189, 204	94
17	3,4-Dimethyl-2-prop-2-enyl-2,5-dihydrothiophene-1,1-dioxide	12.2	0.9	79, 93, 107, 122	80
18	Caryophyllene oxide	14.0	1.4	109, 121, 177, 222	94
	**K1**				
1	4-Carene	4.0	4.3	48, 93, 105, 121	87
2	Acetic acid	9.6	2.9	40, 43, 45, 60	91
3	6-Camphenol	10.0	2.2	67, 93, 108, 119	84
4	α-Bourbonene	10.3	2.4	81, 105, 123, 161	90
5	Bornyl acetate (d)	10.9	6.7	43, 95, 121, 136	94
6	β-Elemene	11.0	9.0	81, 147, 189, 204	92
7	Caryophyllene	11.0	1.3	91, 119, 133, 175	92
*8*	*p*-Menth-1-en-8-ol	11.9	2.3	93, 107, 121, 136	93
9	Verbenone (e)	11.9	19.2	107, 122, 135, 150	94
10	β-Selinene	12.1	18.0	105, 175, 189, 204	94
11	Germacrene D	12.3	3.5	105, 119, 133, 204	87
12	γ-Terpinen	13.6	3.6	77, 90, 195, 136	86
13	Caryophyllene oxide	14.0	5.7	109, 121, 177, 222	93
14	3-Ethyl-3-hydroxyandrostan-17-one	14.1	3.5	91, 172, 190, 218	82
15	Aromadendrene oxide	14.3	2.7	79, 93, 109, 220	83
16	Unknown 1	15.9	4.7	79, 93, 107, 153	-
17	Tetrachlorohydroquinone dimethyl ether	16.5	2.4	181, 209, 261, 276	80
	**S1**				
1	Sabinene (a)	4.2	8.2	41, 77, 93, 136	94
2	7-Εxoethenylbicyclo[4.2.0]oct-1-ene	4.3	2.9	77, 90, 119, 134	89
3	Limonene (b)	6.2	7.9	68, 93, 107, 121	94
4	Eucalyptol	6.4	0.8	43, 81, 108, 154	93
5	2,6-Dimethyl-1,3,5,6-octatraene	6.6	1.5	91, 105, 119, 134	93
6	*p*-Cymene (c)	7.5	4.7	65, 91, 119, 134	94
7	1-Methoxy-1,3,5-cycloheptatriene	9.2	0.8	77, 91, 107, 122	90
8	6-Camphenol	10.1	1.9	67, 93, 108, 119	90
9	α-Bourbonene	10.3	2.1	81, 105, 123, 161	87
10	Bornyl acetate (d)	10.9	3.6	43, 95, 121, 136	91
11	β-Elemene	11.0	4.0	81, 147, 189, 204	94
12	Verbenone (e)	12.0	14.9	107, 122, 135, 150	94
13	β-Selinene	12.1	8.3	105, 175, 189, 204	92
14	*cis*-Verbenol	12.2	7.8	94, 109, 119, 137	93
15	*cis*-Verbenol	12.4	2.5	94, 109, 119, 137	84
16	γ-Terpinene	13.6	1.8	77, 90, 195, 136	87
17	Caryophyllene oxide	14.0	1.8	109, 121, 177, 222	95
18	β-Eudesmol (h)	15.7	3.8	59, 149, 164, 189	94
19	Unknown 1	15.9	1.8	79, 93, 107, 153	-
20	Nerolidol isobutyrate	18.4	7.4	71, 153, 203, 221	81
21	Andrographolide	18.7	1.3	119, 159, 173, 187	83
22	(3*E*,5*E*,7*E*)-6-Methyl-8-(2,6,6-trimethyl-1-cyclohexenyl)-3,5,7-octatrien-2-one	19.5	1.9	109, 159, 243, 258	83

**Table 2 molecules-30-00499-t002:** Compounds detected in *B. papyrifera* resin extracts.

	Compound Name	R.T. (min)	Area (%)	Major Ions (*m/z*)	Similarity Index
	**M2**				
1	Limonene (b)	6.2	14.5	68, 93, 107, 121	94
2	4-Carene	7.0	0.6	48, 93, 105, 121	83
3	Ocimene	7.3	2.1	79, 93, 121, 136	94
4	Perillene	8.1	0.9	69, 81, 135, 150	82
5	4-Cyclopentylidene-2-butanone	9.2	1.5	67, 95, 110, 122	81
6	Octyl acetate (f)	10.0	60.0	70, 84, 98, 112	97
7	Linalool	10.6	1.0	71, 93, 121, 136	90
8	*trans*-Nerolidol (g)	14.4	11.8	69, 107, 161, 189	95
9	β-Eudesmol (h)	15.7	0.8	81, 147, 189, 204	90
	**K2**				
1	Limonene (b)	5.9	0.3	68, 93, 107, 121	93
2	Octyl acetate (f)	9.9	6.9	70, 84, 98, 112	97
*3*	*trans*-Nerolidol (g)	14.4	15.3	69, 107, 161, 189	96
4	Unknown 2	15.8	1.8	41, 161, 187, 229	-
5	Isoaromadendrene epoxide	15.9	1.0	79, 93, 107, 153	86
6	4-Methylene-1-methyl-2-(2-methyl-1-propen-1-yl)-1-vinylcycloheptane	16.1	8.5	50, 107, 161, 204	89
7	Germacrene B	16.2	1.9	93, 121, 161, 204	87
8	Dodecanoic acid	17.0	1.8	78, 115, 157, 200	92
9	12-(Acetoxy)-2,6,10-trimethyl-(*E*,*E*,*E*)-2,6,10-dodecatrien-1-ol	17.8	20.1	93, 135, 150, 218	80
10	Nerolidol isobutyrate	18.4	35.7	71, 153, 203, 221	82
11	Unknown 3	19.2	1.0	73, 121, 129, 145	-
	**S2**				
1	Octyl acetate (f)	10.0	2.1	70, 84, 98, 112	96
*2*	*trans*-Nerolidol (g)	14.4	4.7	69, 107, 161, 189	96
3	β-Eudesmol (h)	15.7	1.8	81, 147, 189, 204	89
4	3-Ethyl-3-hydroxyandrostan-17-one	15.8	0.6	91, 121, 175, 189	89
5	4-Methylene-1-methyl-2-(2-methyl-1-propen-1-yl)-1-vinylcycloheptane	16.1	1.4	50, 107, 161, 204	86
6	Germacrene B	16.2	1.7	93, 121, 161, 204	86
7	Verticiol	16.4	1.2	133, 229, 257, 272	90
8	Dodecanoic acid	17.0	1.6	78, 115, 157, 200	95
9	12-(Acetoxy)-2,6,10-trimethyl-(*E*,*E*,*E*)-2,6,10-dodecatrien-1-ol	17.8	7.8	93, 135, 150, 218	80
10	3-Hydroxy-(3β,17β)-spiro(androst-5-ene-17,1-cyclobutan)-2-one	18.2	2.6	91, 107, 159, 271	80
11	Nerolidol isobutyrate	18.4	11.2	71, 153, 203, 221	83
12	4-Methylene-1-methyl-2-(2-methyl-1-propen-1-yl)-1-vinylcycloheptane	18.7	4.6	50, 107, 161, 204	85
13	Palmitic acid	19.3	2.0	60, 129, 213, 256	91

**Table 3 molecules-30-00499-t003:** Compounds detected in *B. papyrifera* bark extracts.

	Compound	R.T. (min)	Area (%)	Major Ions (*m/z*)	Similarity Index
	**M3**				
1	Limonene (b)	6.2	5.5	68, 93, 107, 121	94
2	Unknown 4	9.2	0.9	67, 95, 110, 122	-
3	Octyl acetate (f)	10.0	72.9	70, 84, 98, 112	97
4	Linalool	10.6	1.0	71, 93, 121, 136	90
5	Geranyl acetate	12.3	0.9	69, 93, 121, 136	93
6	*trans*-Nerolidol (g)	14.4	8.2	69, 107, 161, 189	95
	**K3**				
1	Furfural	9.8	2.3	39, 67, 95, 96	92
2	Octyl acetate (f)	10.0	3.4	70, 84, 98, 112	95
3	*trans-*Nerolidol (g)	14.4	13.4	69, 107, 161, 189	93
4	β-Eudesmol (h)	15.7	2.5	81, 147, 189, 204	91
5	(3β,17β)-3-Hydroxyspiro(androst-5-ene-17,1-cyclobutan)-2-one	15.8	1.8	73, 134, 159, 271	81
6	4-Methylene-1-methyl-2-(2-methyl-1-propen-1-yl)-1-vinylcycloheptane	16.1	0.5	50, 107, 161, 204	81
7	Dodecanoic acid	17.0	2.8	78, 115, 157, 200	89
8	12-(Acetoxy)-2,6,10-trimethyl-2,6,10-(*E*,*E*,*E*)-dodecatrien-1-ol	17.8	27.5	93, 135, 150, 218	81
9	Nerolidol isobutyrate	18.4	26.7	71, 153, 203, 221	82
10	Palmitic acid	19.3	6.9	60, 129, 213, 256	82
	**S3**				
1	Acetic acid	9.6	0.5	40, 43, 45, 60	96
2	Octyl acetate (f)	10.0	2.2	70, 84, 98, 112	96
3	*trans*-Nerolidol (g)	14.4	1.1	69, 107, 161, 189	95
4	β-Eudesmol	15.7	0.6	81, 147, 189, 204	91
5	4-Methylene-1-methyl-2-(2-methyl-1-propen-1-yl)-1-vinylcycloheptane	16.1	0.6	50, 107, 161, 204	88
6	Germacrene B	16.2	0.7	93, 121, 161, 204	85
7	Dodecanoic acid	17.0	0.8	78, 115, 157, 200	91
8	12-(Acetoxy)-2,6,10-trimethyl-2,6,10-(*E*,*E*,*E*)-dodecatrien-1-ol	17.8	9.8	93, 135, 150, 218	80
9	Unknown 5	18.1	1.1	73, 134, 159, 271	-
10	Nerolidol isobutyrate	18.4	7.3	71, 153, 203, 221	83
11	Thunbergol	18.7	1.8	50, 107, 161, 204	85
12	Palmitic acid	19.3	1.8	60, 129, 213, 256	90

## Data Availability

The original contributions presented in this study are included in the article/Appendix A. Further inquiries can be directed to the corresponding author.

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
