# Peer review of "Exploring Volatiles and Biological Effects of Commiphora africana and Boswellia papyrifera Incense"

_molecules, 2025, doi:10.3390/molecules30030499_

Round 1
Reviewer 1 Report
Comments and Suggestions for Authors
In this manuscript, the authors employ gas chromatography-mass spectrometry (GC-MS) to conduct an exhaustive comparative analysis of extracts from the Sudanese medicinal plants C. africana and B. papyrifera. Three distinct sample preparation techniques are utilized to isolate low molecular weight compounds from the samples, with GC-MS playing a pivotal role in elucidating the chemical structures of these compounds. Subsequent cellular assays and antimicrobial experiments were performed to evaluate the biological activities of the extracted fractions. The findings reveal that both extracts exert potent inhibitory effects on cancer cell proliferation and exhibit notable antimicrobial properties, indicating robust potential for therapeutic applications. This research significantly advances the understanding and exploration of natural medicines. The manuscript is well-organized, logically coherent, and the data presented are both precise and credible. Nonetheless, the discussion section falls short in delving into the mechanisms driving these experimental outcomes. The manuscript can be accepted after adding discussions on relevant mechanisms.
Reviewer 2 Report
Comments and Suggestions for Authors
The paper titled “Exploring volatiles and biological effects of Commiphora africana and Boswellia papyrifera incense” is devoted to studying the traditional uses and health benefits of two resins and bark from these plants, which are significant in Sudanese culture for their aromatic properties and medicinal applications. Three different extraction methods were employed, and the resulting extracts were used in the research. The authors determined the composition of the extracts and their biological activities. The findings support the potential therapeutic applications of C. africana and B. papyrifera due to their bioactive compounds.
However, the authors should clarify some points in the article:
1. The authors make some inaccuracies. Rome is not a country (line 88).
2. The authors make some unscientific assumptions when it comes “….to the psychological conditions believed to be caused by the evil eye….”.
3. In the introduction, it would be worthwhile to describe more about what is currently known about the chemical composition of these plants.
4. Do the authors have any information on the differences in the composition of resin and bark of these plants depending on the place of their growth?
5. It needs to be clarified why the authors used only K1-K3 and S1-S3 extracts for biological activity studies. Why did the authors not use extracts M1-M3, mainly since they differed in composition, especially the resin and bark of B. papyrifera?
3. Why were such extract concentrations chosen to study tyrosinase inhibitory activity? The same applies to the concentrations of compounds detected in the extracts.
4. Subparagraphs 4.5 and 4.6 are identical.
5. The authors should describe the statistical analysis of the data in more detail, as well as the repeatability of the experiments.
6. In the article, the authors state that the studied extracts have antiproliferative properties on certain cell lines but do not disclose the molecular mechanisms of their action. It is the understanding of these mechanisms that allows us to assert their therapeutic potential.
Reviewer 3 Report
Comments and Suggestions for Authors
Abstraction:
The entire section is somewhat unclear. For instance, while three extraction methods were employed, the main findings from these methods are not mentioned. The authors mention the function of C. africana extract obtained through smoke solvent trapping; however, there is no information provided regarding the functions of other extracts or those derived from different extraction methods. Authors should succinctly and concisely summarize these main results. Some small issues:
1. English writing, line 21-22
2. line 26-27, competitive inhibitors what do you mean?
3. line 27-32, how about the function of B. papyrifera extract?
Introduction:
The author acknowledged many efficacy of herbal extracts/incense in various traditional medicines; however, the research focus of this article was not adequately emphasized. Some aspects such as differences in extraction methods, variations in active ingredients, roles of tyrosinase and Streptococcus mutans Clarke etc. were only briefly mentioned in the last paragraph, which is insufficient for a comprehensive understanding.
Results:
Line 103-107, can be put in figure captions
Discussion
The author has introduced several extract components, tyrosinase and various inhibitors etc. These elements appear to be more effectively presented in the introduction section. Furthermore, the author should consider comparing the differences between their research results and those of other studies.
Conclusion:
Perhaps I did not fully grasp the author's research intentions; therefore, I feel that the conclusion merely offers a general overview of the research process without adequately capturing the core aspects of the study.
Reviewer 4 Report
Comments and Suggestions for Authors
1- The introduction should be improved with providing the link between the different biological activities assessed and why they are assessed.
2- The number of unknown components in each sample is too high preventing the overview to identify the components, the difference between the various extraction methods, and later understand the component-biological activity links.
3- It would be valuable to check the activity of the main components for antimicrobial activity just as was assessed for the anti tyrosinase one.
4- The discussion is to be enriched, valuable results are reported but not properly discussed.
Round 2
Reviewer 3 Report
Comments and Suggestions for Authors
Abstract: The abstract needs a lot improvement. The background is too verbose, and the purpose is absent. Line 24 “The chromatographic profiles differed ……”This sentence is not appropriate in describing the results. I get what you're saying, but the author's point isn't clear in the abstract. Honestly, I can't figure out what they mean just from this abstraction. Are you looking to compare three extraction methods or are you focusing on the differences in active substances?
The remaining sections are well described.
Reviewer 4 Report
Comments and Suggestions for Authors
Authors made substantial work to improve the manuscript.
Author Response
Thank you for critical reading of our manuscript. The indications were very helpful to improve the quality of our paper. I hope that this version of manuscript is acceptable for the publication in Molecules.